# PET-PZT Dielectric Polarization: Electricity Harvested from Photon Energy

**DOI:** 10.3390/mi15121505

**Published:** 2024-12-18

**Authors:** Alex Nikolov, Sohail Murad, Jongju Lee

**Affiliations:** Department of Chemical and Biological Engineering, Illinois Institute of Technology, Chicago, IL 60616, USA; smurad1@iit.edu (S.M.); jlee210@iit.edu (J.L.)

**Keywords:** ferroelectric actuator, photon–heat–stress, electricity convertor, photon–heat electricity capacitance

## Abstract

The effect of residual stress or heat on ferroelectrics used to convert photons into electricity was investigated. The data analysis reveals that when the PET-PZT piezoelectric transducer is UV-irradiated with a 405 nm wavelength, it becomes a photon–heat–stress electric energy converter and capacitator. Our objective was to evaluate the PET-PZT photon–heat–stress electric energy conversion performance and the role of the light’s wavelength and intensity. Converting waste energy from energy-intensive processes and systems is crucial to reducing the environmental impact and achieving net-zero emissions. To achieve these, innovative materials are needed to efficiently convert ambient energy into electricity through various physical mechanisms, such as the photovoltaic effect, thermoelectricity, piezoelectricity, triboelectricity, and radiofrequency wireless power transfer.

## 1. Introduction

Pyroelectricity is the spontaneous electric response of a polar, dielectric material to a change in temperature. The use of ferroelectrics for harvesting solar energy is an inspiring research area, and this approach has many potential technological applications, such as in pyroelectric actuating devices, light–heat rotor engines, pyroelectric heat converters, motion sensors, infrared thermometers, pyrolytic nuclear fusion (Naranjo et al. [1]), and heat energy harvesting devices (Poplavko and Yakymenko [2]). Pyroelectrics and piezoelectrics are ferroelectric materials: ceramics, minerals, and polymers with a Perovskite molecular crystallographic cubic crystal structure and general formula of ABX_3_ (e.g., LaYbO_3_, CaTiO_3_, and Pb(Zr_0.52_Ti_0.48_)O_3_) that is polarized under external energy impulses and converts energy to voltage. PZT-based-ceramics are some of the most studied ferroelectric materials, and they have a wide field of applications (Pilon and McKinley [3], Mitofsky [4], and Pecunia [5]). The study of ferroelectricity has a long history (Haertling [6] and Lang [7]). For example, tourmaline and aluminum gallium nitride materials exhibit piezoelectricity. Schmidt (1707) first confirmed the pyroelectric phenomenon, and Brewster [8] named it. The temperature heat or cool time variation ∆T/∆t causes normal compressive strain and electrical field polarization that is proportional to the displacement dD, and D = p_c_ dT, where p_c_ is the ferroelectric polarization coefficient. This equation was applied for small displacements, where the charge density Q varies linearly with the temperature variation ∆T as follows: Q = p_c_ ∆T. The ferroelectric force is as follows: F_e_ = p_c_/ε_0_ ε_r_, where ε_0_ and ε_r_ are the permittivity coefficients in a vacuum and the dielectric medium, respectively. The electrical current I_p_, generated by ferroelectric materials during heating or cooling, is on the order of nA or pA. For small amounts of heat–time stress, dT/dt, the current estimation is given by the relation I_p_ = A p_c_ dT/dt, where A is the sample area. The heat–voltage conversion performance requires a consideration of the friction energy loss. Theories of electrical polarization and relaxation materials were investigated by Maxwell [9], Böttcher [10], and Poplavko and Yakymenko [2]. The theory of dipole fluctuation in the nonlinear instability of isotropic dielectrics was developed by Landau and Lifshitz [11] and Pilon and McKinley [3]. Pyroelectric heat energy conversion evaluated with the solid-state thermodynamic cycles is of the order of nW/cm^2^ or nJ/cm^2^ sec. Raman spectroscopy (RNS) was used to characterize the ferroelectric structure, domain texture, and crystallographic misalignments, and it is a powerful tool for studying the crystal orientation and molecular/nanoparticle mechanical stress–dipole polarization. It can also be used to optimize the dielectric polarization electricity performance [12]. Researchers attempted to develop efficient photon–heat–stress electrics conversion materials with a high thermo-conductivity (Lheritier et al. [13], Zhou et al. [14], Mohammadnia and Rezania [15], Wang et al. [16], and Fang et al. [17]). The pyroelectric material serves as an alternate of the solar cell light and allows for the heat–electrical energy conversion. Due to the narrow electron gap (1.1 eV is the forbidden energy gap), piezoelectric materials are not useful as solar energy convertors [18]. The Ni-doping strategy for band gap engineering in developing a photo-responsive piezoelectric was used successfully in a laboratory; however, the technology is not ready to be applied to real-life situations. Here, we will demonstrate an alternative approach for ferroelectric materials’ photon energy conversion. Recently, Wang and Wu [19] discussed the energy output conversion of a pyroelectric device vs. the temperature–time gradient (dT/dt) for different temperatures. Ferroelectric materials have a low heat transfer coefficient and produce a significant gradient (dT/dt) to allow for a heat–electrical conversion. Light interacts with matter in a number of ways: it can be absorbed, reflected, refracted, or emitted. This interaction is the subject of QED (Quantum Electrodynamics) [20]. The electrons of matter absorb the energy of photons, and the energy state of the matter changes, the temperature of the matter increases, the temperature gradient triggers a mechanical stress, and the molecular structure becomes polarized and converts it to electricity. The pyroelectric phenomenon is an electric spontaneous vector polarized under the external heat impulse. The heat impulse triggers mechanical stress, and the mechanical stress is converted to voltage. In the piezoelectric material, the light impulse harvests the heat–mechanical stress and converts it to voltage. The piezoelectric response under the light impulse behaves as a pyroelectric material. In summary, piezoelectric and pyroelectric materials irradiated with light are heat–energy converters. The energy of the light vs. the frequency is given by the Planck equation: E = hν, where h is the Planck constant, and ν is the light frequency. When the piezoelectric or the pyroelectric (ferroelectric) materials are used as light–heat energy converters, the roles of the light wavelength, profile shape (such as square, sinusoidal, or triangular), and intensity regarding the heat energy conversion performance require consideration. Here, we discuss the PET-PZT response to the photon–heat–stress, and the activation of an electric conversion using static and dynamic UV light irradiation.

The ferroelectric materials Pb(Zr_1−*x*_Ti*_x_*)O_3_ (PZT) producing PZT film are market-available, have wide applications, and are extensively used as various sensors, ultrasonic transducers, resonators, and piezoelectric transformers [6,21,22]. In the past decade, PZT films have been recognized as promising for the conversion of energy storage, capacitor materials nonvolatile and dynamic random-access memories, electro-optic modulators, infrared detectors, and micro-electromechanical systems [23,24,25,26].

Converting energy from energy-exhaustive processes and systems is crucial to reducing the environmental impact and achieving net-zero emissions [27]. To achieve these, innovative materials are needed to efficiently convert ambient energy into electricity through various physical mechanisms, such as the photovoltaic effect, thermoelectricity, piezoelectricity, triboelectricity, and radiofrequency wireless power transfer.

## 2. Experimental

This study used a commercial piezoelectric transducer (PET), which is a PZT piezoelectric (Pb^2+^ O^2−^ Ti^4+^ Zr) layer between two discs. These are brass/silver electrodes, and under light–heat–stress, they become an electricity convertor and capacitor. The edge of the brass disc electrode was sealed onto the glass plate with a hole in the center (Figure 1A). The PET irradiation light used was a UV LED 405 ± 5 nm light. The light illumination was normal in relation to the disc’s surface (Figure 1A). The UV 405 nm light source was continuous (nonstop), and there was a strobe light with two frequencies: 2.3 Hz and 7.2 Hz. The light was powered with two parallel 3.7 V, 7.4 Ah lithium batteries (LGDAHA 11865 (LG Chem, Ochang, Cheongju, Republic of Korea)). The UV LED 405 nm light power of 1.4 mW/cm^2^ was measured using a GENERAL UV513AB light meter (General Tools & Instruments LLC, Secaucus, NJ, USA) with a MidOpt Bi405 violet interference band pass filter (Midwest Optical Systems, Inc., Palatine, IL, USA). The distance between the UV head light and disc illumination was 4 cm. The illumination area was 2.5 cm^2^. The UV PET irradiation was conducted in an open area.

The disc’s vertical deflection (bent) was monitored using a 560 nm collimated 0.08 mm laser beam reflection from a 5 mm glass right-angle prism placed on the disc surface’s edge (Figure 1A). The right-angle glass prism had a broadband antireflection (BBAR) coating, a product of Edmund Optics, Barrington, NJ, USA.

The glass prism was placed on the brass disc surface’s edge (Figure 1A). The laser incident beam was reflected from the prism’s bottom surface, and the vertical disc deflection movement was monitored. The light reflected from the prism was picked up by the phototransistor (TIL 99 NPN IR (Texas Instruments, Dallas, TX, USA)). The phototransistor was connected to a DC linear millivolt amplifier. The distance between the glass prism and phototransistor sensor was 50 cm, and the signal was picked up by the digital oscilloscope. The PET–PZT brass electrode was UV-405 nm-light-illuminated and its voltage response was monitored by the oscilloscope. The brass disc was UV-405 nm-illuminated, and its temperature vs. time was monitored. The UV 405 nm light power of 1.4 mW/cm^2^ was used to illuminate the PET-PZT and brass electrode. The PET’s positive silver electrode was electrically shielded. The PET-PZT was connected to the oscilloscope with a coaxial cable via a BNC connector (Figure 1A,B).

## 3. Results

The data for the PET-PZT and PET brass disc electrode voltage response to the 405-nm light photon energy during the illumination time in the voltage picked up by the oscilloscope are presented in Figure 2A. The PET-PZT brass electrode voltage response to the light–heat (resulting in volume expansion–contraction) can be seen in the violet curve in Figure 2A. The curve has two branches. The left branch of the curve depicts the start of the light illumination of the thermal heating. During the UV 405 nm heating, the mV increases with time until it reaches its optimal value, when the light heating flux becomes equal to the air convection cooling flux after 110 s (at room temperature, 22.4 °C). The shape of the light on the heating curve fits a second-order polynomial equation. When the UV 405 nm light was turned off, the air convection cooling began. The voltage decayed exponentially until it reached 250 s, which was when it reached its original room temperature value. The light heating and air convection cooling curves have different time trends because they are related to different phenomena. The PET-PZT UV irradiation voltage time response shown in the yellow curve in Figure 2A follows the light’s start and stop, and it has two upside-down branches. When the light was turned on, the PET’s quick response was 75 mV (the positive voltage jump); then, the voltage followed the decay curve, and after 110 s, it reached the zero voltage. When the light was turned off (−75 mV), the voltage jump response appeared, and the voltage time trend increased to zero. The output voltage vs. time, shown in the yellow curve in Figure 2A, matched the data presented in Figure 6.1 (as reported by Knopf and Uchino [28] in Ch. 6) and was obtained using PLZT.

The branches have the same shape (see Figure 2A, the curve in yellow). The curves’ shape is explained in the discussion part of this paper.

The brass disc electrode response to the UV 405 nm light conditions of “on” for heating and “off” for air convection cooling vs. time was measured with a type K thermocouple, a product of Fisher Scientific. The temperature-sensitive part (the thermocouple) was fixed on the central area of the brass disc in a small pit in contact with the brass disc. The central part of the disc area (an area of 2.4 cm^2^) was irradiated with UV 405 nm light. The thermocouple wires were connected to the millivolt section of a calibrated Fluke 87 V thermocouple. The data for the temperature vs. time are presented in Figure 2B. The graph in Figure 2B depicts the PET brass disc electrode response to the UV 405 nm light heating “on” and “off”, along with the air convection cooling vs. time. After 110 s, the temperature increased by 6.9 °C and reached its optimum temperature. The shape of the temperature heating curve (light on) fits a second-order polynomial equation. The curve light off (air convection cooling) decays exponentially, and after 250 s, it reached its original room temperature value of 22.4 °C. The similarity of the shape vs. time of the violet curve in Figure 2A and the voltage vs. time and the temperature vs. time in Figure 2B is due to the brass disc electrode volume expansion and contraction effect.

The PET UV 405 nm light response to the frequency of the light illumination was studied. The UV 405 nm strobe light with the square-shaped signal with a frequency of 2.3 Hz was applied. The signal pick-up by the phototransistor, a fast response device, was observed on the oscilloscope (Figure 3A). The PET-PZT shape response signal observed on the oscilloscope is different from what is shown in Figure 3A. The PET-PZT pick-up signal had the same frequency as the phototransistor of 2.3 Hz. The PET response signal shape had the saw-blade shape shown in Figure 3B, rather than the square shape. Why do the two signals have the same frequency but different shapes? The frequency and shape of the voltage response signal provide information about how the device interacts with the photons. Two different devices picked up the signals in Figure 3A,B. The phototransistor (a fast response device) is a device that operates by directly converting photons to electrons and electricity. The PET-PZT (piezoelectric actuator) is a device that operates by spontaneous molecular polarization activated by mechanical stress to produce electricity. When photons interact with the piezoelectric material, they produce mechanical stress to activate the molecular dipole polarization, and then electricity is generated. This is a different phenomenon than the photons-to-electrons conversion. The data presented in Figure 3A,B provide proof of how the PET-PZT light was converted to piezoelectricity. Most of the piezoelectric materials do not interact with light, since their energy gap (band gap) is beyond the photon energies of the visible light [18,29]. The PET-PZT light photon response signal picked up by the oscilloscope is shown in Figure 3B.

The PET-PZT time response vs. illuminated light frequency was tested with a 7.2 Hz square-shaped signal. The PET-PZT response was monitored by an oscilloscope (Figure 4A,B). It is interesting that the shape of the curve presented in Figure 4B was like the curve shape presented in Figure 2C published by Chen et al. [30] using a PLZT ceramic.

The shape of the response signal in Figure 4B of the PET-UV 405 nm when illuminated with the frequency of the 7.2 Hz square-shaped signal was analogous to the data presented in Figure 2A (violet curve). The data in Figure 3B and Figure 4B have two parts: on the left side, the light was on and the voltage increases with the heating time, and on the right side, the voltage decreases with the cooling time. The curved shape during light heating time fits with a second-order polynomial equation. The curved shape during the air convection cooling time when the light was off fits with an exponential decay equation. The response signals’ shape vs. frequency requires an explanation, which is presented in the discussion part.

The pyroelectricity effect is usually temperature change vs. time (dT/dt) and is triggred by the dipole moment change in a unit volume of polar materials [31,32]. The pyroelectricity effect has being composed of “Heat harvesting technologies capture of the waste heat conversion through the thermodynamic heat engine across various working media” [27].

## 4. Discussion

The data presented in Figure 2A,B, Figure 3B and Figure 4B require an explanation for how the photon-thermal energy was converted to electric energy. Let us analyze the shape of the curve of the brass disc that is irradiated with UV 405 nm light and its temperature vs. time (Figure 2B).

### 4.1. Photon Energy Thermal Effect Evaluation

How photons interact with matter was briefly discussed in the introduction. Here, we discuss the photon heat convention phenomenon. When the electrons of matter absorb the photon and the matter’s energy state increases, one option is that the temperature of the matter increases. As an example, the photon energy is absorbed by matter, as when the brass disc volume expands during the heating time. The brass disc’s mass was 2.4 g, and the brass disc’s temperature increased by 1 °C (the disc diameter was 3.5 cm; its thickness was 0.03 cm; and the brass density was 8.4 g/cm^3^). The specific heat of the brass was 0.4 J/g °C. The light source power was 970 mW per cm^2^ (970/9.6 ≈ 100 mW/cm^2^) or 100 mJ/sec.cm^2^, which was needed to increase the temperature of the plate by 1 °C. A single photon of energy from the 405 nm light was evaluated using Planck’s energy equation E = hν, where h is Planck’s constant, and ν is the light wave frequency; it was 4.8 × 10^−19^ J. The measured light source power was 1.4 mW/cm^2^ and 1.4 mJ/s·cm^2^. The estimated photon flux density was (4.8 × 10^−19^ J) × (2.9 × 10^15^) ≈ 1.4 mJ/s·cm^2^. The mean photon flux density was 4 × 10^15^ (photons/s·cm^2^ with 1.4 mW and a wavelength of 405 nm). The photon flux with a density of 1.4 mJ/s cm^2^ and 70 s of light illumination time was required to heat the brass plate by 1 °C. The beginning of the heating process (light on) using the data presented in Figure 2B shows the temperature rate ≈ 0.02 °C/s. As the light heats the last brass disc plate, the temperature increases. The temperature difference ∆T between the brass disc and the surrounding air increases, and the air convection cooling flux increases. Over time, the photon heating flux and the air convection cooling flux become equal, and the brass disc’s temperature stops increasing. When the light was turned off, the air convection cooling flux reduced the disc temperature, and after 150 s, the disc temperature reached the ambient temperature. Why do the curves in Figure 2A (violet curve) and Figure 2B have identical shapes? The data presented in Figure 2A,B depict the brass disc response during the light on and off time periods. The data in Figure 2A’s violet curve indicated a thermo-mechanical response: the expansion–contraction response of the brass disc during light on and off time periods. The data presented in Figure 2B show the temperature of the brass disc with the “light on” heating and the “light off”, and includes the air convection cooling vs. time. Both curves in Figure 2A,B depict the brass disc photon thermo effect vs. time. Figure 2A’s violet curve shows the photon-thermo-mechanical strain (expansion–contraction effect), and Figure 2B shows the brass disc’s photon-thermo-temperature effect. The data for Figure 2A’s yellow curve depict the piezoelectricity activated by the photon thermo-mechanical strain. The data in Figure 2A (the yellow curve) will be examined next, and we will provide an explanation of the curve shape.

### 4.2. Dielectric Polarization Electricity Harvesting from Photon Energy

The piezoelectricity was activated when the mechanical stress was applied. When the light was turned on, the PET-PZT brass electrode absorbed the photon energy, and this was converted to heat. The heat-produced strain triggered the piezoelectricity, and a 75 mV voltage jump appears in the voltage vs. time curve marked in yellow in Figure 2A. The energy was converted to electrical energy, the electrodes become charged, and the electrical energy was stored (capacitance). As the light irradiation continued, the brass electrode temperature increased and the temperature gradient slowly decreased due to the air convection cooling (see Figure 2B). The temperature gradient was not strong enough to activate the stress and charge. The electrodes’ charge over time slowly discharged to zero (Figure 2A, the yellow curve). The slope trend of the two curves, the PET-PZT electrodes’ potential (Figure 2A, yellow curve), and the PET-PZT brass electrode thermo-strain (Figure 2A, violet curve) correlated with our explanation. When the electrodes’ potential became zero (Figure 2A, yellow curve) and the light was turned off, the temperature dropped quickly due to the air convection cooling and activated the thermo-stress (Figure 2B). The −75 mV voltage jump appears in the voltage vs. time curve marked in yellow in Figure 2A. This observation reveals that the heating or cooling stress activates dielectric polarization electricity and the PET-PZT electrodes’ charge–discharge process, respectively.

### 4.3. Photon Energy-PET-PZT Dielectric Polarization Electricity: Conversion Evaluation

The data analysis for the yellow curve in Figure 2A reveals that when the light was on, a voltage jump with the amplitude of 75 mV occurred. The surface charge on the electrodes after 50 s diminishes to zero (Figure 2A, yellow curve). The light was turned on and off to activate the photon–electron surface change and convert it to electrical energy. This approach creates a great opportunity and requires additional study. The scope of future work will include a study of the PET-PZT photon–heat–stress electrical charge activation and its optimization. We demonstrated that photon–heat–stress electric energy conversion required heating or cooling stress on the PET-PZT brass electrode. The way to activate the heat–cooling stress was to irradiate the PET-PZT using a strobe light. The data presented in Figure 3B and Figure 4B depict the PET-PZT response signal using a 2.3 Hz and a 7.2 Hz square-shaped signal to simulate the light being on and off over time. The shape of the PET-PZT response signal under the light frequency in Figure 3B and Figure 4B had the saw-blade shape and was correlated to the heating and air convection cooling data presented in Figure 2B. As the frequency increases from 2.3 Hz to 7.2 Hz, the photon-heating time decreases, and so the air convection cooling time also decreases. When the heating time decreases, the heat stress decreases and the photon–heat energy conversion to electricity is less. The way to increases, the photon–heat–stress electric energy conversion is to use high-intensity light irradiation at an optimal frequency. The two light parameters (frequency and intensity) needed to optimize the PZT photon–heat–electric conversion require an experimental study. The investigation of the crystallographic orientation and the residual stress field in ferroelectrics will use Raman spectroscopy to study the dielectric polarization and heat transfer coefficient as factors in enhancing the efficacy of the photon–heat–electricity conversion. Recent studies have shown an increasing output density from 10 to 10^3^ µW/m^2^ [27].

## 5. Conclusions

We demonstrated how the PET-PZT (piezoelectric actuator) irradiated with UV405 nm light activates photon–heat–stress, polarizes the dielectric material, and converts the energy into electricity.

Improving the performance of the PET-PZT photon–heat–stress energy conversion to electricity requires the optimization of the light frequency and intensity.

The data presented in Figure 2A (yellow curve) reveal the photon–stress electricity conversion and show that the PET-PZT operates as an electric capacitor.

## Figures and Tables

**Figure 1 micromachines-15-01505-f001:**
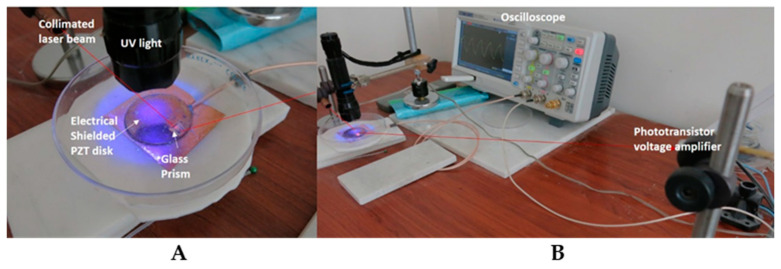
The experimental set-up to study the PET-UV 405 nm light response. (**A**). UV 405 nm light, PET-PZT disc, and right-angle glass prism. (**B**). Red laser beam, laser beam pick-up stand with phototransistor voltage amplifier, and digital oscilloscope.

**Figure 2 micromachines-15-01505-f002:**
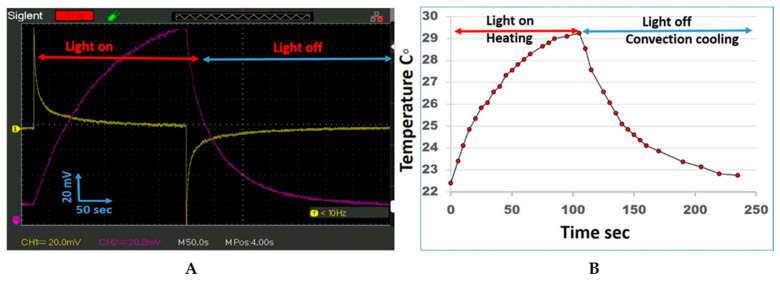
(**A**). Oscilloscope data for the PET-PZT and PET brass disc electrode voltage response to the UV 405 nm light irradiation during the on and off periods. The PET-PZT data are shown in yellow, and the PET brass disc electrode data are shown in violet. The red and blue horizontal lines on the figure depict the UV 405 nm light illumination time periods (on and off). (**B**). The brass disc response to the UV 405 nm light heating on, temperature rising and light off due to air convection cooling, and temperature decreasing.

**Figure 3 micromachines-15-01505-f003:**
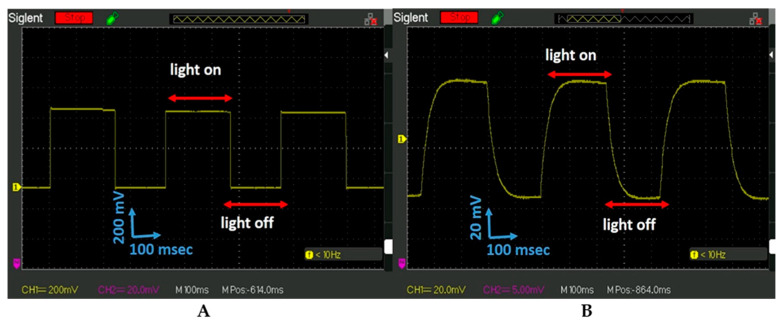
PET light response signal picked up by the oscilloscope under UVA 405 nm light illumination: (**A**). The UVA 405 nm 2.3 Hz square-shaped signal picked up by the phototransistor (a fast response unit). (**B**). UVA 405 nm 2.3 Hz square-shape signal picked up by the PET-PZT unit and observed by the oscilloscope.

**Figure 4 micromachines-15-01505-f004:**
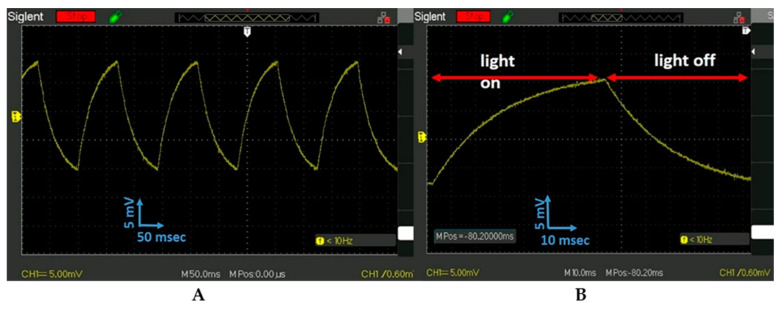
(**A**). The response signal of PET-PZT on the oscilloscope when it was illuminated with the UV 405 nm 7.2 Hz square signal. (**B**). The signal of the single unit in (**A**) was expanded five times to see the effect of the light being on and off.

## Data Availability

The data presented in this study are available on request from the corresponding author.

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
