# Peer review of "PET-PZT Dielectric Polarization: Electricity Harvested from Photon Energy"

_micromachines, 2024, doi:10.3390/mi15121505_

Round 1
Reviewer 1 Report
Comments and Suggestions for Authors
1. Page 1, line 37. p_c should be the pyroelectric coefficient rather than the ferroelectric polarization coefficient.
2. Page 2, line 67. The definition of QED is suggested to be given.
3. Page 2, line 72. There may be some unclear discussion. Normally, the pyroelectricity is composed of four significant parts (doi.org/10.1038/s42254-024-00771-8), i.e., primary pyroelectricity (spontaneous polarization changes with respect to temporal temperature variations), secondary pyroelectricity (piezoelectric effect due to thermomechanical strain/expansion), tertiary pyroelectricity (flexoelectric effect due to spatial strain gradient and deformation that ascribed from non-uniform heating or cooling), and field-induced pyroelectricity (temperature dependence of dielectric polarization under an external electric field). The aforementioned “heat impulse triggers mechanical stress, and the mechanical stress is converted to voltage” is suggested to be rewritten accordingly.
4. The statement “The piezoelectric response under the light impulse behaves as a pyroelectric material” could be reasonable in terms of crystal classes. However, the phenomena of how light triggers piezoelectric to generate voltage could be different and is recommended to be discussed further (doi.org/10.1093/nsr/nwad186).
5. What are the wavelength selectivity and sensitivity of PET-PZT materials under light stimuli?
Author Response
Reviewer’s comment:
- Page 1, line 37. p_c should be the pyroelectric coefficient rather than the ferroelectric polarization coefficient.
Reply:
The term ferroelectric polarization coefficient was changed to pyroelectric coefficients.
Reviewer’s comment:
- Page 2, line 67. The definition of QED is suggested to be given.
Reply:
The term QED was changed to Quantum Electrodynamics.
Reviewer’s comment:
- Page 2, line 72. There may be some unclear discussion. Normally, the pyroelectricity is composed of four significant parts (doi.org/10.1038/s42254-024-00771-8), i.e., primary pyroelectricity (spontaneous polarization changes with respect to temporal temperature variations), secondary pyroelectricity (piezoelectric effect due to thermomechanical strain/expansion), tertiary pyroelectricity (flexoelectric effect due to spatial strain gradient and deformation that ascribed from non-uniform heating or cooling), and field-induced pyroelectricity (temperature dependence of dielectric polarization under an external electric field). The aforementioned “heat impulse triggers mechanical stress, and the mechanical stress is converted to voltage” is suggested to be rewritten accordingly.
Reply:
Following the discussion of the pyroelectricity four composed parts in the recently published article in Zhou et al. [1] “Heat harvesting technologies capture this waste heat through thermodynamic heat engine across various working media” in our manuscript the role of the thermomechanical strain/expansion was the dominated effect and was discussed in the revised manuscript.
Reviewer’s comment:
- The statement “The piezoelectric response under the light impulse behaves as a pyroelectric material” could be reasonable in terms of crystal classes. However, the phenomena of how light triggers piezoelectric to generate voltage could be different and is recommended to be discussed further (doi.org/10.1093/nsr/nwad186).
Reply:
How the light triggers the piezoelectric to generate voltages in the case of Ti-rich film of Pb(Zr Ti )O (PZT)1−x x 3 R(LT) at Ti/Zr ratio of 0.5, in the case of monoclinic crystal phase, was discussed in the manuscript. The manuscript data resented in figures 2,3 and 4 reveals the domination of the thermomechanical strain/expansion. The tetragonal crystal structure occurs when the ratio Ti/Zr is increased. The increase of Ti atoms in the PET-PZT (Pb(Zr1−xTix)O3 (PZT) film tetragonal crystal structure, enhances the ferroelectric performance respectively the conversion efficiency also increases. The reference “Non-planar dielectrics derived thermal and electrostatic filed inhomogeneity for boosted weather-adaptive energy harvesting.”[2] In the revised manuscript was considered.
Reviewer’s comment:
- What are the wavelength selectivity and sensitivity of PET-PZT materials under light stimuli?
Reply:
The used wavelength was 405±5 nm. The red light of 650 nm was explored. The intensity of the two light sources was not compatible, and it was not possible to evaluate the sensitivity of PET-PZT sensor. To evaluate the PZT wavelength sensitivity and performance requires evaluation of the light wavelength absorption and reflectivity of the PET-PZT brass electrode.
References
- Zhou, Y.; Ding, T.; Xu, G.; Yang, S.; Qiu, C.-W.; He, J.; Ho, G.W. Sustainable heat harvesting via thermal nonlinearity. Nature Reviews Physics 2024, 1-15.
- Zhou, Y.; Ding, T.; Cheng, Y.; Huang, Y.; Wang, W.; Yang, J.; Xie, L.; Ho, G.W.; He, J. Non-planar dielectrics derived thermal and electrostatic field inhomogeneity for boosted weather-adaptive energy harvesting. National Science Review 2023, 10, nwad186.

Reviewer 2 Report
Comments and Suggestions for Authors
1. The authors have claimed that “Due to the narrow electron gap (1.1 eV is the forbidden energy gap), piezoelectric materials are not useful as solar energy convertors”. However, this rule have some still applicable cases for example reported elsewhere 10.1016/j.mattod.2023.05.026, which can be mentioned in the manuscript. Also, the most of piezoelectric materials have a higher bandgap that the authors have mentioned.
2. In addition, no bandgap measurements were provided in the manuscript. In case no data a re available provide strong evidence citing references.
3. Generally, mechanical strain produced by deformation and thus direct piezoelectric effect allows generating electricity. In case of heat, before any strain is generated, some heat dissipation may be expected due to impurities, defects etc. It is also wavelength dependent process. In this respect the efficiency of photon-driven stress can not be efficient compared with mechanically-driven strain (deformation). Also, provide some details on wavelength optimization strategies to be used to maximize the electric output and efficiency.
4. In conclusion the authors have stated that the developed materials behave as a capacitor, however, in this case the bandgap should be higher than the authors have reported or efficiency of charge storage will be reduced due to leakage currents.
5. I would suggest some discussion on solar light irradiation, how effective would it be for photon-driven stress formation and its effect on electricity generation mechanisms.
6. The authors have used relatively low frequencies of the light irradiation. What happens in case the frequency will be higher. Would the studied processes be more efficient and to what extend in case it can be estimated quantitatively?
Author Response
Reviewer’s Comment:
- The authors have claimed that “Due to the narrow electron gap (1.1 eV is the forbidden energy gap), piezoelectric materials are not useful as solar energy convertors”. However, this rule has some still applicable cases for example reported elsewhere 10.1016/j.mattod.2023.05.026, which can be mentioned in the manuscript. Also, the most of piezoelectric materials have a higher bandgap that the authors have mentioned.
Reply:
The bang gap of the Pb(Zr Ti )O (PZT)1−x x 3 R(LT) Ti/Zr, at ratio of 0.5,and the monoclinic crystal is 3.5 eV (see Samanta et al. [1] Band gap and, piezoelectricity and temperature dependance of differential permittivity and energy storage of PZT with different Zr/Ti ratios, Vacuum, 156,(2018)456-462.)
Reviewer’s Comment:
- In addition, no bandgap measurements were provided in the manuscript. In case no data are available provide strong evidence citing references.
Reply:
Please see the reply 1 to the Reviewer’s comments.
Reviewer’s Comment:
- Generally, mechanical strain produced by deformation and thus direct piezoelectric effect allows generating electricity. In case of heat, before any strain is generated, some heat dissipation may be expected due to impurities, defects etc. It is also wavelength dependent process. In this respect the efficiency of photon-driven stress cannot be efficient compared with mechanically-driven strain (deformation). Also, provide some details on wavelength optimization strategies to be used to maximize the electric output and efficiency.
Reply:
How the light triggers the piezoelectric to generate voltages in the case of Ti-rich film of Pb(Zr Ti )O (PZT)1−x x 3 R(LT) at Ti/Zr ratio of 0.5, in the case of monoclinic crystal phase, was discussed in the manuscript. The manuscript data resented in figures 2,3 and 4 reveals the domination of the thermomechanical strain/expansion. The tetragonal crystal structure dominates when the ratio Ti/Zr is increased with the increase of Ti atoms. The PET-PZT (Pb(Zr1−xTix)O3 (PZT) film tetragonal crystal dominance at Ti/Zr ratio > 0.5 and the ferroelectric performance respectively the conversion efficiency increases. The reference (see Zhou et al. [2] “Non-planar dielectrics derived thermal and electrostatic filed inhomogeneity for boosted weather-adaptive energy harvesting”) In the revised manuscript this was discussed. The piezoelectric to generate voltage is wavelength sensitive. The thermos-sensitivity at 405±5 nm was studied (see the manuscript fig 2). The red light of 650 nm was also tested. The source intensity was not compatible, and it was not capable of evaluating the sensitivity of PZT film vs. wavelength. The light wavelength reflection and absorption of the brass electrode of the PET-PZT requires examination when evaluating the wavelength sensitivity.
Reviewer’s Comment:
- In conclusion the authors have stated that the developed materials behave as a capacitor, however, in this case the bandgap should be higher than the authors have reported or efficiency of charge storage will be reduced due to leakage currents.
Reply:
The wave gap of PEZ is 3.5 eV (see the reply 1). The PEZ film is between two metal electrodes brass and silver), and is of 35 nF capacitor. The capacitor is light sensitive and because of light radiation heath the dielectric constant is temperature sensitive.
Reviewer’s Comment:
- I would suggest some discussion on solar light irradiation, how effective would it be for photon-driven stress formation and its effect on electricity generation mechanisms.
Reply:
The PET-PEZ is a solar radiation -voltage sensitive.
Reviewer’s Comment:
- The authors have used relatively low frequencies of the light irradiation. What happens in case the frequency will be higher. Would the studied processes be more efficient and to what extend in case it can be estimated quantitatively?
Reply:
The figures 3 (2.3 Hz) and 4 (7.2 Hz) in the manuscript reveals that the amplitude of the oscillatory signal vs. frequency is decreased because the heating time decreases.
References
- Samanta, S.; Sankaranarayanan, V.; Sethupathi, K. Band gap, piezoelectricity and temperature dependence of differential permittivity and energy storage density of PZT with different Zr/Ti ratios. Vacuum 2018, 156, 456-462.
- Zhou, Y.; Ding, T.; Cheng, Y.; Huang, Y.; Wang, W.; Yang, J.; Xie, L.; Ho, G.W.; He, J. Non-planar dielectrics derived thermal and electrostatic field inhomogeneity for boosted weather-adaptive energy harvesting. National Science Review 2023, 10, nwad186.
